# Early Detection and Monitoring of Gastrointestinal Infections Using Syndromic Surveillance: A Systematic Review

**DOI:** 10.3390/ijerph21040489

**Published:** 2024-04-17

**Authors:** Olubusola Adedire, Nicola K. Love, Helen E. Hughes, Iain Buchan, Roberto Vivancos, Alex J. Elliot

**Affiliations:** 1Institute of Population Health, University of Liverpool, Liverpool L69 3GF, UK; buchan@liverpool.ac.uk; 2Real-Time Syndromic Surveillance Team, Field Services, Health Protection Operations, UK Health Security Agency, Birmingham B2 4BH, UK; helen.hughes@ukhsa.gov.uk (H.E.H.); alex.elliot@ukhsa.gov.uk (A.J.E.); 3National Institute for Health Research Health Protection Research Unit in Gastrointestinal Infections, University of Liverpool, Liverpool L69 7BE, UK; nicola.love@ukhsa.gov.uk (N.K.L.); roberto.vivancos@ukhsa.gov.uk (R.V.); 4Institute of Infection, Veterinary and Ecological Sciences, University of Liverpool, Wirral CH64 7TE, UK; 5Field Services North-West, Health Protection Operations, UK Health Security Agency, Liverpool L3 1DS, UK

**Keywords:** gastrointestinal infections, early detection, diarrhoea, vomiting, gastroenteritis, syndromic surveillance

## Abstract

The underreporting of laboratory-reported cases of community-based gastrointestinal (GI) infections poses a challenge for epidemiologists understanding the burden and seasonal patterns of GI pathogens. Syndromic surveillance has the potential to overcome the limitations of laboratory reporting through real-time data and more representative population coverage. This systematic review summarizes the utility of syndromic surveillance for early detection and surveillance of GI infections. Relevant articles were identified using the following keyword combinations: ‘early warning’, ‘detection’, ‘gastrointestinal activity’, ‘gastrointestinal infections’, ‘syndrome monitoring’, ‘real-time monitoring’, ‘syndromic surveillance’. In total, 1820 studies were identified, 126 duplicates were removed, and 1694 studies were reviewed. Data extraction focused on studies reporting the routine use and effectiveness of syndromic surveillance for GI infections using relevant GI symptoms. Eligible studies (*n* = 29) were included in the narrative synthesis. Syndromic surveillance for GI infections has been implemented and validated for routine use in ten countries, with emergency department attendances being the most common source. Evidence suggests that syndromic surveillance can be effective in the early detection and routine monitoring of GI infections; however, 24% of the included studies did not provide conclusive findings. Further investigation is necessary to comprehensively understand the strengths and limitations associated with each type of syndromic surveillance system.

## 1. Introduction

The global burden of gastrointestinal (GI) infections combines the impact of incidence, severity, and duration. Understanding this burden is essential for tackling the threat GI infections pose to public health and allocating resources and efforts accordingly. However, there are several challenges to achieving this understanding, such as inconsistent definitions of disease and symptoms, difficulties in identifying the causative agent for many cases, underreporting of cases to health authorities, and incompatible reporting systems.

Globally, diarrhoeal diseases continue to pose significant health challenges across all ages. In 2016, it was estimated that there were 1.7 million deaths attributable to diarrhoea among all ages, while there were nearly 500,000 deaths estimated in children aged under 5 years [1]. Despite the number of deaths in children decreasing over the last two decades, diarrhoeal diseases are still the fifth leading cause of death in young children with a significant burden in Sub-Saharan Africa [1]. It is, therefore, crucial to acknowledge that GI infections represent a significant public health concern, extending beyond mild GI symptoms, and have a substantial impact on humanity’s wellbeing.

GI infections encompass a wide range of conditions affecting the entire GI tract [2]. Acute GI illnesses (including gastroenteritis) are often characterized by often sudden onset and short-lived self-limiting symptoms including diarrhoea, vomiting, nausea and abdominal discomfort, cramps, and bloody diarrhoea.

Traditional public health surveillance methods for reporting GI infections typically rely on laboratory diagnosis of specific GI pathogens [3]. However, the commonly presenting symptoms of gastroenteritis, such as diarrhoea and vomiting, are often self-limiting and non-severe; therefore, patients do not present to healthcare services and subsequent microbiological testing of clinical samples is not undertaken frequently. For example, the second Infectious Intestinal Disease Study (IID2) of England illustrated that for every national surveillance case of norovirus, there were 13 GP consultations and 288 community cases [4]. The development of an alternative type of surveillance over the last three decades has the potential to significantly improve the detection of GI disease in the community (both outbreaks and seasonal increases in activity). This approach, known as ‘syndromic surveillance’, involves the ongoing and systematic collection, analysis, and interpretation of symptom-specific data for the early identification of adverse disease events [5]. In addition to facilitating the timely detection and reporting of monitored diseases, syndromic surveillance can offer ad hoc services such as situational awareness and event characterization.

While there is substantial evidence of the utility of syndromic surveillance for monitoring respiratory diseases in the community, such as influenza and respiratory syncytial virus [6], the utility of syndromic surveillance in GI infection surveillance is less clear. There is a smaller evidence base for its utility, and some of the published literature presents conflicting evidence [7,8]. Here, we report a systematic review which identifies and describes evidence of the utility of syndromic surveillance for the early detection and monitoring of GI infections. The results of this review will add to the existing evidence base and indicate priorities for development of effective GI syndromic surveillance.

## 2. Materials and Methods

This systematic review was carried out following the Preferred Reporting Items for Systematic Reviews and Meta-Analyses (PRISMA) guidelines and was registered on Prospero, reference number CRD42022321839 [9].

A systematic search of the literature was conducted using PubMed, Medline, Scopus and CINAHL (Cumulative Index to Nursing and Allied Health Literature) online databases. Search terms were generated to identify papers, published in English, indicating an operational syndromic surveillance system that collect, analyze, and report in a real-time system for routine use. It is also important that these surveillance systems are validated and can drive public health action. The papers required the inclusion of keywords relating to both syndromic surveillance AND gastrointestinal infection in the title and/or the abstract (Appendix A).

We incorporated studies that investigated GI infections by utilizing syndromes associated with GI symptoms including vomiting, diarrhoea, nausea, and abdominal pain. The search was set from 2000 to March 2024, focusing exclusively on peer-reviewed studies published in the English language. The inclusion criteria required that studies employed syndromic surveillance for routine public health surveillance purposes, and the system had been approved for continuous routine use. Additionally, syndromic systems that involved the monitoring of over-the-counter medication usage for GI infections and the assessment of school absenteeism related to GI illnesses were also considered. Where systematic reviews were identified during the search process, we reviewed the sources cited within each review to identify any original research studies that described a relevant system which was not identified in our initial search.

We excluded studies that did not involve the use of real-time, syndromic surveillance systems for collecting and analyzing health-related data. Traditional laboratory-based surveillance, event-based surveillance for specific events, non-GI symptoms or diseases, simulation-based studies, and analyses related to temporary emergency circumstances or natural disasters were therefore not considered. We focused on peer-reviewed journal articles written in English and did not include sources such as book chapters, dissertations, technical reports, conference abstracts, letters, interviews, case reports, and systematic reviews. Outbreaks originating from contaminated food and water sources were also excluded from our analysis.

Our decision to limit the search to English-language, peer-reviewed publications may have introduced bias. It is also possible that syndromic surveillance systems not covered by published literature have been excluded.

The process of selecting studies for inclusion in our analysis was conducted by two independent reviewers (OA and NL) using Covidence [10]. Initially, all titles and abstracts were screened to identify studies that either reported on, or seemed to report on, syndromic surveillance of GI infections. Following this, the researchers (O.A. and H.E.H.) conducted a comprehensive evaluation of the complete texts to identify the chosen studies that fulfilled the predefined inclusion criteria. Any discrepancies were resolved by a third reviewer (A.J.E.).

Data were extracted using a standardized data collection tool developed in Microsoft Excel 2024. The data extracted from the studies included: system name, system country, system description, purpose of the system, and system start date. Where available, information was also extracted describing the technical details of the system such as timing, frequency and methods of data collection and transfer to the syndromic surveillance database, sensitivity, and specificity of system. Qualitative details included symptoms of interest, seasonality of GI infection, patient and public involvement, detection of GI infections, utility of the syndromic surveillance system, underlying data infrastructure, and public health benefits of the system. The primary reviewer (O.A.) conducted data extraction from all studies, while a secondary reviewer (H.E.H.) performed a quality control check by extracting information from a randomly selected 50% sample of studies.

In conducting this systematic review, a rigorous methodological approach was followed to ensure the inclusion of high-quality qualitative studies. The Joanna Briggs Institute (JBI) checklist for qualitative studies was utilized as a framework for assessing the methodological rigor and relevance of the included studies [11]. The JBI checklist was selected for use as it was developed by international collaboration of researchers, clinicians, and educators which ensures the checklists were informed by a wide range of perspectives. This provided a systematic and comprehensive set of criteria that guided the evaluation of study design, data collection, analysis, and the overall trustworthiness of the qualitative evidence. By employing the JBI checklist, this review aimed to maintain transparency, minimize bias, and ensure the inclusion of robust qualitative studies to enhance the validity and reliability of the findings.

## 3. Results

In total, 1817 journal articles were identified with publication dates from 2000 to March 2024 (Figure 1). Three additional articles were added by snowballing references from systematic reviews identified in the search (snowballing is a method by which new references were identified from primary studies). Duplicates (*n* = 126) were removed from the 1820 studies. Upon de-duplication, 1694 studies were screened by titles and abstracts with 1665 studies excluded as they did not meet the inclusion criteria. In total, 29 studies were identified as eligible for inclusion in the review. The quality control check performed by the second reviewer did not reveal any major discrepancies.

### 3.1. Summary of the Syndromic Surveillance Systems

All 29 studies selected through the review reported the same main objective: to gather information on a real-time basis that could be used to inform public health action (there were, however, differences in the specific aims of each system, Appendix A). This objective included the use of surveillance systems to track both seasonal and irregular occurrences of GI infections as well as to identify and monitor the increases in GI infections at specific times in the year. The 29 studies identified from the full screen included a range of different syndromic surveillance systems from ten countries and territories across Korea, United States, France, Sweden, Netherlands, Spain, Canada, United Kingdom, Japan, and Portugal. Just over half of the studies (52%, 15/29) were reported from the United States and United Kingdom [8,12,13,14,15,16,17,18,19,20,21,22,23,24,25]. About 55% (16/29) [13,16,22,23,24,25,26,27,28,29,30,31,32,33,34,35] of the studies indicated that syndromic surveillance exhibited the potential to detect GI infections earlier than traditional surveillance, while a further 10% (3/29) of studies indicated that syndromic surveillance exhibits the potential to detect GI infections or symptoms at an early stage but with limitations [15,20,36]; a further 3% (1/29) highlighted that syndromic surveillance exhibits potential to detect GI infections or symptoms at an early stage only when combined with traditional surveillance system [18].

The usefulness of syndromic surveillance in identifying GI infections or symptoms at an early stage was deemed unclear in 24% (7/29) of the included studies [8,12,17,19,21,37,38], while 2/29 of the included studies suggested that syndromic surveillance was incapable of detecting GI infections at an earlier stage in comparison to traditional surveillance [14,39].

The use of syndromic surveillance systems for continuous day to day monitoring of GI infections to either track seasonal peaks, surges, or irregular occurrences of GI infections were explicitly stated in 90% (27/29) of the included studies [8,12,13,15,16,18,19,20,21,22,23,24,25,26,27,28,29,30,31,32,33,34,35,36,37,38,39].

### 3.2. Type of Syndromic Surveillance Systems

The syndromic surveillance systems reported in the selected studies described different sources of syndromic data, of which the emergency department (ED) setting was the most frequently utilized [12,15,16,17,19,24,27,33,34,36]. Of the included studies, 3/29 reported the use of multiple or combined syndromic surveillance systems [24,25,27]. The other systems reported in the included studies were National Health Service (NHS) Direct calls (a telehealth system) [13], GP surveillance [26,29,38], integrated medical records from inpatient, outpatient, laboratory, and pharmacy data [18,21,23,24,25,27,28,35,39], electronic medical records [8,18,23,32], a nurse advice hotline [21], nursing home public health surveillance system [28], internet search queries [30], and nursery school absenteeism [31].

### 3.3. Quality of Included Studies

The included studies were critically appraised [11] as: high quality, 15/19 [12,13,22,23,24,25,28,29,30,31,32,33,34,36,37]; medium quality, 10/29 [14,15,16,17,18,21,27,35,38,39]; or low quality 4/29 [8,19,20,26] (Appendix A).

### 3.4. Patient Public Involvement Reporting

None of the 29 studies reported any patient or public involvement in designing the studies, choice of systems, or reporting the results.

## 4. Discussion

### 4.1. Main Findings of This Study

This review has provided an oversight of the literature describing approaches to undertaking syndromic surveillance of GI infections. A key finding from this review has been the heterogeneity of syndromic surveillance systems in use for GI infection surveillance. While ED surveillance was the most common syndromic data source, other sources included telehealth calls, GP attendances, electronic medical records, ambulatory electronic records, nursing advice hotline, internet-based surveillance, children day care centres, nursery school surveillance, and pharmacy sales [14,18,20,28,30,31,35,37,39]. Across the different types of syndromic surveillance systems, symptoms commonly linked to GI symptoms included acute gastroenteritis, diarrhoea, and vomiting. However, this review observed different outcomes and conclusions about the degrees of effectiveness of syndromic surveillance systems for GI surveillance.

### 4.2. How Does This Compare to Others?

The use of syndromic surveillance has become widespread to identify and monitor infectious and non-communicable diseases. Detecting changes in activity in a timely manner can help improve the understanding of burden of the diseases, trigger public health action and give public reassurance. Ahn et al. reported an ED syndromic surveillance program developed by The Korean Centre for Disease Control and Prevention, which assisted the EDs to detect early increases in the number of target GI symptoms, e.g., acute diarrhoea, acute rash symptoms, and acute haemorrhagic fever symptoms visiting the emergency room [36]. Bounoure et al. provided further evidence on the use of syndromic surveillance to capture the geographical spread of GI disease, reporting that electronic medical record surveillance gave a daily count of medicalized acute gastroenteritis at the municipal level [8]. Similarly, Smith et al. reported that GP syndromic surveillance produced timely data on GI illness at a local level, while also linking prescriptions of GI medications to morbidity [14].

This review also revealed the ability of syndromic surveillance to monitor GI symptoms in specific populations and settings. Delespierre et al. reported how a national ecological nursing home public health surveillance system aided early detection and monitoring of acute gastroenteritis in the elderly residents [28]. Enserink et al. demonstrated circulating viruses and parasites, rather than bacteria, contribute to seasonal gastroenteritis experienced by children in Dutch day care centres [37]. The use of a school absenteeism surveillance system reported by Tanaba et al. highlighted its practical use as an infection control measure for enterohaemorrhagic Escherichia coli infection, gastroenteritis infection, and in containing symptoms of fever, diarrhoea, and vomiting in nursery and school settings [31].

Syndromic surveillance is intended to enhance and support, not replace, traditional infectious disease surveillance, such as notifiable disease or laboratory-based surveillance. Syndromic surveillance should provide a timely alert before other surveillance systems detects an outbreak or seasonal increases in activity thereby enabling more timely investigation and more rapid implementation of interventions. An analysis of nurse advice hotline data suggested it was 4–50 h timelier for symptom detection than outpatient office visit data [21]. Similarly, Donaldson et al. reported that NHS 111 calls for diarrhoea provided a two-week lead time ahead of NHS 111 telehealth calls for vomiting [22]. The combination of syndromic and laboratory surveillance can generate hypotheses about the potential aetiology behind the increases in GI activity such as attributing increases in diarrhoea cases found through GP surveillance to either rotavirus or norovirus [38]. Integrated healthcare delivery system utilizing comprehensive electronic medical records with inpatient, outpatient, laboratory, and pharmacy data helped to investigate GI occurrences in specific settings such as day care centres and restaurants, before confirmed laboratory pathogen identification.

Early studies reporting syndromic surveillance had reported the use of syndromic surveillance for detecting ‘bioterrorist’ incidents and securing mass events [16,39]. Syndromic surveillance has also been used for the identification and monitoring of infectious and non-infectious diseases, particularly in identification of seasonal trends of illnesses like influenza [6]. As detailed in Table 1, this review identified the use of different surveillance systems for the early identification, as well as routine monitoring of GI seasonal trends. Edelstein et al. [30] confirmed the use of Websök, an internet-based surveillance system which reflects on-going circulation of norovirus in the community. On average, this syndromic surveillance system preceded norovirus detection by laboratory data (reflecting hospital-based activity) by 2–3 weeks. The added benefit of this system was to support local hospital infection control teams prepare for seasonal norovirus activity and it has since been integrated in routine norovirus surveillance, alongside laboratory surveillance. Similarly, Loveridge et al. [13] indicated in their analysis that telehealth syndromic surveillance can also predict early warning signs of norovirus infection. This study in England issued alerts when 4% or more of telehealth vomiting calls in all age groups were reported for two weeks in a row, with the resulting threshold providing up to four weeks’ advance warning of forthcoming norovirus pressures on the health service [13].

Syndromic surveillance systems are most useful when followed by appropriate public health actions. These actions can come in the form of issuing alerts to healthcare professionals or health services [13], predicting and preparing for seasonal occurrences of infections [30], and infection control by public health advice [31]. In some instances, syndromic surveillance has been used to assess changes in healthcare-seeking behaviours and an indirect method of assessing adherence to infection control measures, particularly during the COVID-19 pandemic [23,24,34]. Existing surveillance systems also provide an avenue to invent newer and better systems. This was highlighted by Bounoure et al., where electronic medical records helped to improve existing infectious disease surveillance systems [8]. Rodriguez et al. further highlighted the potential of ED surveillance to be adapted into designing a system to detect bioterrorism and for surveillance of naturally occurring epidemics [17].

This review provides some evidence that syndromic surveillance systems are effective in routine monitoring of GI infections in specific populations and settings such as nursing homes, children day care centres, and schools. By combining syndromic surveillance with laboratory-based surveillance, researchers can also generate hypotheses about the potential causes of increased GI activity, facilitating targeted investigations and interventions. In addition, it demonstrated the ability to provide early detection of GI infections and seasonal increases in activity thereby enabling timely public health responses. However, almost half of the evidence remains inconclusive to support these claims. The results of our study contrast with a previous systematic review that showed inconclusive evidence of the effectiveness of syndromic surveillance for the early detection of waterborne outbreaks [42]. However, these two reviews likely indicate that syndromic surveillance is more effective for supporting the surveillance of seasonal pathogens at population level rather than for waterborne outbreaks, which are often localized and yield small numbers of cases.

Certain syndromic surveillance systems, such as Websök internet-based surveillance founded on using search queries, and the telehealth surveillance system, have shown their specific timeliness and predictive capabilities in detecting GI infections [30]. These systems provide early warning signs and that can support infection control measures. Moreover, syndromic surveillance can be integrated into routine surveillance practices and public health actions, such as issuing alerts to healthcare professionals, predicting, and preparing for seasonal occurrences, and providing infection control guidance.

The review highlights the potential of leveraging existing surveillance systems, such as electronic medical records and ED surveillance, to improve and develop newer and more effective syndromic surveillance systems for GI infections. As electronic medical and personal health record system data become more routinely linked, curated, and analyzed for population health management, there is also scope for GI surveillance to embed in the artificial intelligence (AI) of such systems. This automation may include interaction with individuals over GI symptoms via personal health record apps in future.

None of the studies we identified reported patient and public involvement. Given that primary care (and increasingly primary care systems incorporating artificial intelligence), are the first points of contact with healthcare services for patients with GI symptoms, it is crucial for research in this field to incorporate the perspectives and experiences of individuals who utilize these services. Furthermore, syndromic surveillance systems, in contrast to laboratory-based systems, heavily depend on the public’s active participation in providing or collecting information. For instance, in cases of school absenteeism monitoring, individuals may not be aware that data from NHS 111 online searches and calls are utilized for this purpose. Therefore, it is crucial to consider engaging the public to gather their perspectives on this matter.

### 4.3. Strengths and Limitations

There were several limitations in this review. First, we only searched and included studies written in English language. We also excluded grey literature and conference abstracts, which contributes to publication bias of the review. Our decision to limit the search to English-language, peer-reviewed publications may have introduced bias. It is also possible that there are syndromic surveillance systems not covered by the published literature that have been excluded. The absence of studies from low- and middle-income countries (LMIC) in the search results highlights the limited development of syndromic surveillance in these regions. It is also important to acknowledge the potential bias resulting from the fact that a significant proportion of the included studies originated from only two countries, France, and the United States. In addition, there is no objective measurement to define utility of syndromic surveillance, as utility was evaluated based on the subjective opinion of the authors of the individual studies included in this review. Overall, the generalizability of the review cannot be ascertained as the methods for implementing syndromic surveillance were disparate across countries and founded on variable underlying healthcare infrastructures. Finally, this review focused on the utility of syndromic surveillance for early detection and surveillance of GI infections, it does not cover important aspects of these systems such as feasibility, implementation, cost/benefit, or operational challenges, which is a limitation. These are important aspects which were not within the scope of our review but could be the focus of future research.

## 5. Conclusions

This review highlights the need for further high-quality studies to generate further evidence on the effectiveness of syndromic surveillance for the early detection and monitoring of GI infections. Further research is required in this area such as developing a comprehensive understanding of the strengths and limitations of each syndromic surveillance system type, or which of the systems is more sensitive to changes in community-based GI activity. Also, there is scope for further exploration of the role of novel surveillance data sources in strengthening GI surveillance, including internet search data or social media trends [43].

This review has highlighted the lack of development of syndromic surveillance in LMIC. France and the United States were the main countries of origin of the syndromic surveillance systems; however, there were no studies that were based in LMICs. Conversely, there is published literature illustrating the use of syndromic surveillance in LMIC for non-GI surveillance, e.g., COVID-19 [44]. Therefore, our study highlights a lack of development of GI-specific syndromic surveillance in LMIC, thereby stressing the need for development in this area. Frameworks and experience developed by countries delivering existing syndromic surveillance systems should be shared with LMIC to expand the use of syndromic surveillance in countries where the health impact of GI infections is more significant; therefore, introducing syndromic surveillance will have a greater impact.

## Figures and Tables

**Figure 1 ijerph-21-00489-f001:**
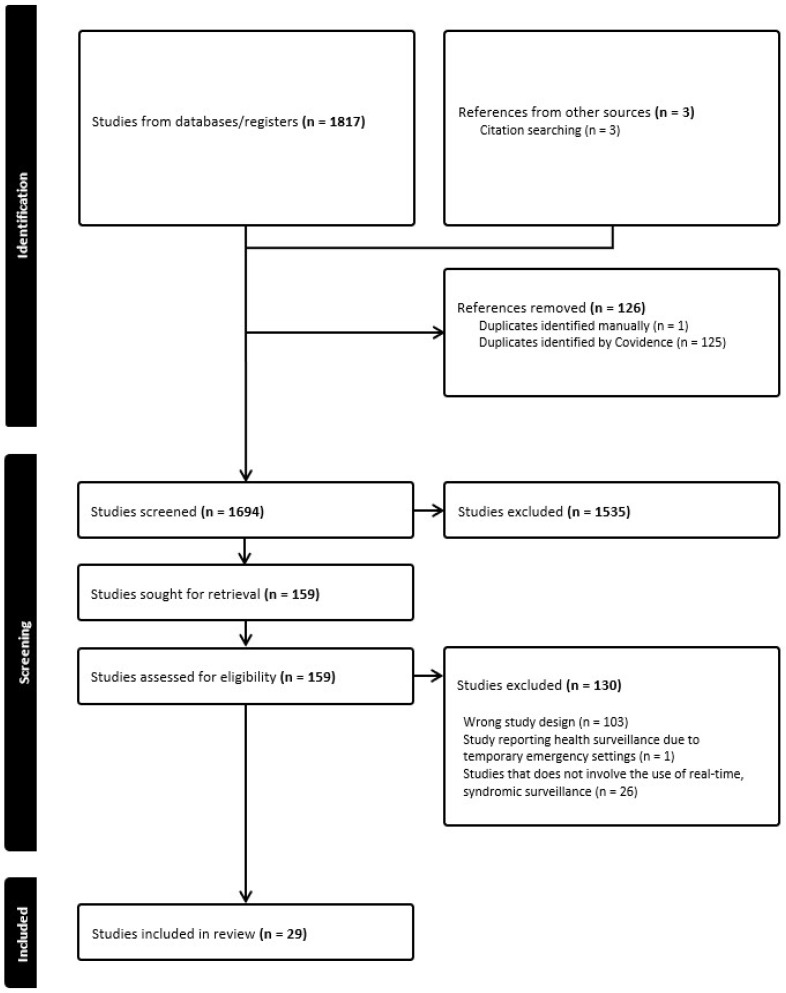
PRISMA flowchart outlining the results of the literature search.

**Table 1 ijerph-21-00489-t001:** Surveillance systems by data source, countries, capabilities, symptoms of GI infection, and patient and public involvement reporting.

Author/Year	Syndromic Surveillance Data Source	Country	Syndromic Surveillance System(s) Capable of Early Detection of GI Infections	GI Infection Symptoms Monitored	PPI ^1^ Reporting	Quality of Included Studies ^2^
Armistead 2022 [23]	Electronic medical record	United States	Yes	Acute gastroenteritis	No	High
Ahn 2010 [36]	ED ^3^	Korea	Yes	Acute diarrhoea, acute rash symptoms, acute haemorrhagic fever symptoms	No	High
Balter 2005 [15]	ED	United States	Yes (with limitations)	Vomiting, diarrhoea, fever	No	Medium
Bounoure 2020 [8]	Electronic medical record	United States	Unclear	Medicalised acute gastroenteritis	No	Low
Brottet 2015 [26]	GP	France	Yes	Gastroenteritis, acute diarrhoea	No	Low
Caillère 2013 [27]	ED, GP	France	Yes	Acute diarrhoea, gastroenteritis	No	Medium
Cho 2021 [32]	Electronic medical record	South Korea	Yes	Diarrhoea, vomiting, abdominal pain, acute gastroenteritis	No	High
Delespierre 2018 [28]	Nursing home surveillance	France	Yes	Acute gastroenteritis	No	High
Donaldson 2022 [22]	NHS 111 calls	United Kingdom	Yes	Gastroenteritis, vomiting, diarrhoea	No	High
Edelstein 2014 [30]	Websök internet-based surveillance	Sweden	Yes	Vomiting, diarrhoea/winter vomiting disease	No	High
Enserink 2015 [40]	Children day care surveillance centres	Netherlands	Unclear	Gastroenteritis	No	High
Flamand 2008 [29]	GP	France	Yes	Gastrointestinal infections (not specified)	No	High
Gerstel 2009 [38]	GP	Spain	Unclear	Diarrhoea	No	Medium
Greene 2012 [18]	Electronic medical Records	United States	Yes (only when combined with traditional surveillance system)	Vomiting, gastroenteritis, nausea, diarrhoea	No	Medium
Heffernan 2004 [16]	ED	United States	Yes	Fever, diarrhoea, and vomiting	No	Medium
Henry 2004 [21]	Nurse advice hotline	United States	Unclear	Fever, gastrointestinal infections (not specified), haemorrhagic	No	Medium
Hripcsak 2009 [20]	Ambulatory electronic records	United States	Yes (with limitations)	Diarrhoea, stomach-ache, vomiting	No	Low
Hughes 2020 [12]	ED	United Kingdom	Unclear	Gastroenteritis	No	High
Kim 2023 [33]	ED	Korea	Yes	Diarrhoea, watery diarrhoea, abdominal pain, fever, nausea, vomiting	No	High
Love 2023 [24]	GP, ED, NHS 111 calls	United Kingdom	Yes	Diarrhoea, vomiting, abdominal pain	No	High
Loveridge 2010 [13]	NHS Direct calls	United Kingdom	Yes	Diarrhoea, vomiting	No	High
Lucaccioni 2021 [35]	Electronic medical record	Portugal	Yes	Acute gastroenteritis	No	Medium
Muchaal 2015 [41]	Pharmacy sales	Canada	No	Acute gastrointestinal illness	No	Medium
Nisavanh 2022 [34]	ED	France	Yes	Acute gastroenteritis, diarrhoea, vomiting, abdominal pain, bloody diarrhoea, fever, nausea, headache	No	High
Olson 2020 [19]	ED	United States	Unclear	Diarrhoea	No	Low
Ondrikova 2023 [25]	GP, NHS 111 calls	United Kingdom	Yes	Vomiting, gastroenteritis	No	High
Rodriguez 2007 [17]	ED	United States	Unclear	Gastroenteritis	No	Medium
Smith 2007 [14]	GP ^4^	United Kingdom	No	Vomiting	No	Medium
Tanabe 2018 [31]	Nursery school surveillance	Japan	Yes	Diarrhoea, vomiting, fever	No	High

^1^ PPI—patient and public involvement; ^2^ quality score assigned using the Joanna Briggs Institute (JBI) checklist; ^3^ ED—emergency department; ^4^ GP—general practitioner.

## Data Availability

Data sharing is not applicable to this article; however, applications for requests to access relevant information included in this study should be submitted to the UKHSA Office for Data Release. Available at: https://www.gov.uk/government/publications/accessing-ukhsa-protected-data (accessed on 30 January 2024).

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
