# Peer review of "Early Detection and Monitoring of Gastrointestinal Infections Using Syndromic Surveillance: A Systematic Review"

_ijerph, 2024, doi:10.3390/ijerph21040489_

Round 1

Reviewer 1 Report

Comments and Suggestions for Authors

This is an interesting study wich could,  however, be improved significantly in terms of design, presentation and the drawing of conclusions. Moreover, it is not very innovative, as such syndromic surveillance are presently being implemented, eg in Greece after the severe floods, and such information is nowhere to be found in the manuscript.

No discrimination is possible through this way of literature search between non-infectious dyspeptic syndromes and infectious causes.

No studies from low-income countries have been included, eg Fekada et al 2022, which are also the countries most severely affected. 

Literature list needs significant updating.

Examples from application of analogous surveillance and epidemics that were identified in this was (eg in England) should be brought up to strengthen the argument.

In general the manuscript is not of significant priority.

Reviewer 2 Report

Comments and Suggestions for Authors

The authors provide extensive insights into the subject. They name the strenghts and limitations of the research clearly. The chapter format is transparant and the review is evolving quite logically.

The final eligible number of studies is very low. It assures that all included studies are of medium to high quality but it is realistic to assume that other studies with interesting but not fully coherent approaches are not included. Did the authors considered checking the outcome of the literature search using less stringent criteria in a random sample?

The subject of feasibility, implementation and cost/benefit of the surveillance systems, although probably very diverse, is not touched? 

The level on which the surveillance is acting is also very different, ranging from institutions (LTCFs, daycare centers,..), to county, region or country. The goals of the surveillance are also variable: early warning, continuous monitoring,... It can be difficult for the reader to estimate the comments on the different systems correctly. An extra figure or a supplementary table, next to Table 1,  could guide the reader herein.

Some abbreviations such as NHS 111 are not clear to non UK residents.

Round 2

Reviewer 1 Report

Comments and Suggestions for Authors

The authors have addressed the points raised adequately.